


# Post-processing of seasonal predictions – Case studies using the EUROSIP hindcast data base

Emmanuel Roulin[1] and Stéphane Vannitsem[1]

[1]Institut Royal Météorologique de Belgique, Avenue Circulaire, 3, 1180 Brussels, Belgium

**Correspondence:** E. Roulin (emmanuel.roulin@meteo.be)

**Abstract.**

Seasonal predictions from climate models are increasingly invoked in various sectors like water management, energy and transport to cite a few. This study investigates the post-processing of the seasonal predictions of the EUROSIP multi-model system. The hindcasts comprise samples of 23 to 36 years and ensembles of 10 to 28 members depending on the 5 models

included. Skill scores both deterministic and probabilistic are calculated in order to compare the impact of the post-processing and help selecting – if any – the multi- or single-model and the post-processing method best suited for a specific location, target season and lead-time. The presence of trends and the cross-validation setting add some complexity to the already heterogeneous database. This study focuses on six cases in Western Europe and the Mediterranean Region. The forecasts of three monthly averages of surface temperature and of mean sea level pressure are compared with the corresponding ERA Interim reanalysis

data whereas the forecasts of precipitation are evaluated with the rain-gauge data from the Global Precipitation Climatology Centre.

The skills of seasonal predictions in the extra-tropics are limited and our results are no exception. There is a significant skill for the spring temperature forecast with model initiation in March for all but one case studies and the skill is extending to the initiation begin of February for Belgium. There is is also a significant skill for the summer temperature for the case studies in

the Mediterranean region. For these area, the skill comes in large part from the global warming so that after having de-trended the data, a null improvement cannot be excluded. Autumn temperature in UK and in the Turkey is forecast with some skill as well as winter temperature in UK and Greece. Precipitation is even more difficult to forecast: the two spots where skill scores are significantly positive are Sweden and Greece during winter with initialisation on the first December. It has been shown that multi-model ensemble improve the skills in many cases and that taking into account the longest common period

of hindcasts results in better and less uncertain skill scores. For all these cases, the post-processing method and the model or model combination resulting in the best skill score have been selected.

## 1 Introduction

Information about the climate during the next seasons is useful in sectors like health, energy, transport, agriculture and water management. In the context of the latter, a dynamical approach consists in initializing a hydrological model with the current

hydrological conditions and forcing this model with calibrated and downscaled seasonal forecasts of atmospheric variables





from coupled atmosphere-ocean general circulation models so that the streamflow can be predicted several months ahead (e.g. Singla et al., 2012; Crochemore et al., 2016; Candogan Yossef et al., 2017; Arnal et al., 2018; Emerton et al., 2018).

The main sources of predictability of the climate at the seasonal scale are found in slow varying boundary conditions like the ocean surface temperature, sea-ice, snow cover and soil moisture. The main driver of the winter climate in Europe is the

North Atlantic Oscillation (NAO) (Doblas-Reyes et al., 2003). NAO may be defined as the difference between the mean sea level pressure at the latitude of the Azores and that in Iceland. When NAO is positive, westerlies are stronger and advection of warm and humid air over Northern Europe is increased. When the NAO is negative, Northern Europe experiences colder temperature and generally drier conditions. Current operational state-of-the-art seasonal prediction systems are able to forecast NAO with skill: e.g. Scaife et al. (2014) reported an anomaly correlation coefficient of 0.62 with GloSea5 on 20-year hindcasts

and Athanasiadis et al. (2017) improved to a value of 0.74 using a multi-model including GloSea5, CFSv2 and CMCC Seasonal Prediction System v1.5. The 110-year hindcast prepared by Weisheimer et al. (2019) with a system similar to the ECMWF S4 resulted in an anomaly correlation of 0.31 and 0.44 on the last 30 years. Ensemble forecasting is a key element in seasonal forecasting as averaging across a large ensemble eliminates significantly the inherently unpredictable weather variability on such long time scales, part of the impact of the model error and so allows the predictable signal to be detected. Despite

these advances the forecasts of near-surface parameters is less successful – for precipitation even less than for temperature. Statistical downscaling techniques allow to achieve better skill (e.g. Baker et al., 2018). Dunstone et al. (2018) also obtained promising results for summer Northern Europe rainfall using very large ensembles (> 80 members). The skill originates from low frequency in North Atlantic sea surface temperature which control moisture availability and hence the convective rainfall efficiency.

Due to large model biases, the added value of long-range forecasts can most often only be seen after forecast calibration (Van Schaeybroeck and Vannitsem, 2018). Much of the litterature on seasonal predictions report skill in terms of anomaly correlation coefficients. This implicitly involves a recalibration of the model output and requires a reference period where to define the observed and model climatologies. Sansom et al. (2016) introduce a recalibration framework that includes adjustments for both unconditional and conditional biases in the mean and variance of the forecast distribution and linear time-dependent bias

in the mean. In their analyse the optimal training period is 30 year for trend-adjusted forecasts and around 15 years otherwise.

The verification of seasonal predictions is also facing the issues of low predictability and small samples (Mason, 2012). Kumar (2009) shows that for seasonal predictions in the extratropical latitudes where the signal to noise ratio is low, and the spread in the skill estimates can be as large as their expected value. Bradley et al. (2008) derive analytical expressions for the uncertainty of Brier Skill Score (BSS) estimate. The approximation underestimates the standard error at small sample sizes (a

few hundred forecast–observation pairs or less) for infrequently occurring events. Non-parametric methods based on bootstrap sampling (Wilks, 2011) are suited to deal with non-Gaussian distributions like those of skill scores and confidence intervals can be corrected for biases in the estimates (Efron and Tibshirani, 1993). Siegert et al. (2016) propose a Bayesian inferential framework to account for uncertainty due to limited numbers of past forecasts and observations. Siegert et al. (2017) use a power analysis framework to estimate the probability of correctly detecting a skill improvement. They show that sample sizes





of climate hindcasts should be increased to about 40 years to ensure sufficiently high power. Wilks (2016) explain how the control of the false discoveries rate allows to interpret multiple tests.

According to Hagedorn et al. (2005), a multi-model ensemble (MME) is superior to single model ensembles (SMEs) not only because error are compensating each other and the diagnostics are non-linear but in particular by its greater consistency
across different aspects of the prediction (season, lead-time, parameter, etc.) and reliability in the long run. A key point in their argumentation is that multi-model should be compared to the average of single models and not to a single best model since a model cannot be the best under all aspects. Similarly, in the DEMETER hindcast database they investigate, there is no model which is consistently bad. Analysing the hindcast data from the Asia-Pacific Economic Cooperation (APEC) Climate Network Yoo and Kang (2005) find that the highest skill can be obtained by selecting several skilful models which are less dependent
each other. Using stochastic models, Weigel and Bowler (2009) show that the net skill improvement due to MME combination is larger if the ensembles to be combined are overconfident. Weigel et al. (2009) note that the joint application of both MME combination and climate conserving recalibration could further optimize the forecasts but only if first the model combination and then the recalibration are applied. Athanasiadis et al. (2017) explain that the high skills in predicting wintertime NAO with a MME are largely due to increasing the ensemble size noting that this effect is expected to be more pronounced when
combining systems with skills that are far from saturation due to limited ensemble size and low signal to noise ratio. High skills are also partly due to increasing model diversity since individual systems may excel in the representation of a different physical process contributing to predictability (Athanasiadis et al., 2017). Methods to combine SMEs vary in complexity from pooling all the members of all SMEs in one single large MME to weighting the SMEs according to their past performances (see Van Schaeybroeck and Vannitsem, 2018, for a review).

Seasonal prediction systems require the setting up of the corresponding database of retrospective forecasts or hindcasts for calibration and skill assessment. These hindcasts vary in sample size (number of years), ensemble size, production mode (fixed or on-the-fly) and other elements. These characteristics add challenges for their use in the multi-model context. The North American Multi-Model Ensemble (NMME) includes many models which are versions of similar models. Slater et al. (2017) propose to reduce the biases arising from models performing similarly with a method based on the principal components
analysis before conducting the Bayesian updating. The hindcasts of the different models included in the European operational multi-model Seasonal to Inter-annual Prediction system – (EUROSIP, Stockdale et al. (2009)) have different numbers of years. In their assessment of the skill of EUROSIP, Mishra et al. (2018) select a common available period of 1992-2012. As we have seen that sample length and ensemble size are critical, we take these two parameters as integral parts of a seasonal prediction system and we propose to take advantage of the full hindcast information and evaluate each model on the longest series
available and pool the ensembles together, each bringing all its members.

In this work, we make use of the EUROSIP hindcast database to identify for 6 areas, cases (season, lead time) where the seasonal predictions near surface temperature, precipitation and mean sea level pressure have skill. The post-processing (recalibration) is applied explicitly as a distinct step. Different post-processing methods are compared. Single models and multi-models combinations are evaluated each over the maximum hindcast length available. The databases and the methodology are
explained in Section 2, summary tables are presented and discussed in Section 3. Conclusions are drawn in Section 4.





**Table 1.** Composition of the EUROSIP hindcast data base; (T) label for the range of years, (N) sample size, (M) ensemble size; GLOSEA5 in single model ensemble M=40 (hindcasts S12 and S13), in multi-model ensemble M=28 (hindcasts S13 only); multi-models: ({(A),–,C,...}) such that A may be included or not, B is excluded and C is required; in total 26 combinations, T=b,...,h, M=22–90.

| Centre | System | Short | Sample | $T$ | $N$ | $M$ | Reference |
|--------|--------|-------|--------|-----|-----|-----|-----------|
| ECMWF | SEAS5 | A | 1981-2016 | $a$ | 36 | 25 | Stockdale et al. (2018) |
| UKMO | GLOSEA5 | B | 1993-2015 | $b$ | 23 | 40/28 | MacLachlan et al. (2015) |
| MF | ARPEGE 5 | C | 1991-2014 | $c$ | 24 | 15 | Voldoire et al. (2013) |
| NCEP | CFSv2 | D | 1982-2010 | $d$ | 29 | 12 | Saha et al. (2014) |
| JMA | JMA/MRI-CPS2 | E | 1981-2015 | $e$ | 35 | 10 | Takaya et al. (2018) |
| Multi-model | {(A),B,(C),D,(E)} | | 1993-2010 | $f$ | 18 | | |
| | {(A),B,C,–,(E)} | | 1993-2014 | $g$ | 22 | | |
| | {(A),–,C,D,(E)} | | 1991-2010 | $h$ | 20 | | |

## 2 Material and method

### 2.1 Hindcast and observation data

The seasonal forecasts have been retrieved from the EUROSIP hindcast database as it was in 2018. Data from the 5 participating centres differ, among other, in the sample ranges of years and the ensemble size (Table 1). For simplicity, the models have been

5  named "A" to "E". For the verification of the single model ensembles (SME), the full sample of each model is used, respectively "$a$" to "$e$". The hindcasts of the GLOSEA5 system (model "B") are produced "on-the-fly" and have been renewed several times since GLOSEA5 is operational. The last two versions (named system 12 and system 13 in the EUROSIP database and hereafter S12 and S13) cover the same period 1993-2015 and have been merged in this study for the skill assessment of single model ensembles. For the study of multi-model ensemble, only the hindcasts S13 have been used.

10  This study is motivated by the development of seasonal hydrological outlooks for two area: the first area includes the Scheldt and Meuse river basins in Belgium and upstream in France and secondly, the second area is the Aliakmon river basin in Northern Greece. This investigation on the seasonal prediction of atmospheric variable focuses on six domains of $2.5° \times 5.0°$ along two transects in Western Europe and the Mediterranean Region (Fig 1) centred on the target basins. The four seasons are considered (spring or "MAM", etc.) and forecasts with lead times 0, 1 and 2 months are evaluated. The hindcasts ensembles

15  pooled and averaged over three months and over the domains are referred to as the hindcasts.

The hindcasts of temperature and of mean sea level pressure are compared to the ERA-Interim reanalysis data (Dee et al., 2011) and the hindcasts of precipitation to the Global Precipitation Centre monthly data (Schneider et al., 2018). These verification data are also averaged over the seasons and over the case studies area and as such are taken as the "observations".



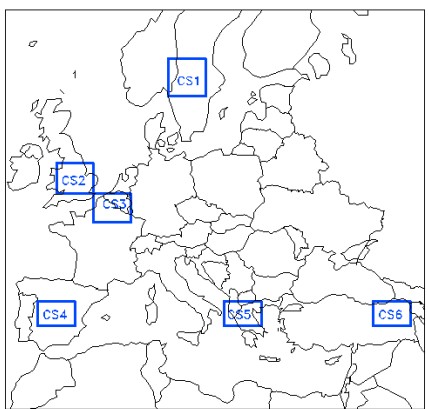

**Figure 1.** Case studies.

## 2.2 Post-processing of temperature and mean sea level pressure

The seasonal predictions of temperature and the mean sea level pressure are corrected according to a statistical model adjusting separately the ensemble mean $\overline{F}_t$ and the spread of the deviation to the mean $e_t^m$ of the ensemble members $m$:

$$F_{Ct}^m = \alpha + \beta \overline{F}_t + \gamma e_t^m \tag{1}$$

5     Three methods are compared:

- A simple bias correction (in short "bc") is applied: $\beta = \gamma = 1$ and $\alpha = \langle X_t - \overline{F}_{Ht} \rangle_t$ i.e. the average difference between observations and hindcast ensemble means.

- The variance inflation ("infl") addresses the correction of biases in the climatological mean and variance and the ensemble reliability (Johnson and Bowler, 2009; Wood and Schaake, 2008):

$$\beta = R_{X\overline{F}_H} \frac{S_X}{S_{\overline{F}_H}}, \alpha = \langle X_t \rangle_t - \beta \langle \overline{F}_{Ht} \rangle_t, \gamma = \sqrt{\left(1 - R_{X\overline{F}_H}^2\right)} \frac{S_X}{S_e}. \tag{2}$$

where $S_X$ and $S_{\overline{F}_H}$ are the standard deviations of the observations and the ensemble means, respectively, $R_{X\overline{F}_H}$ is the Pearson correlation coefficient between both series, and where $S_e^2 = \left\langle S_{F_H^m}^2 \right\rangle_t$ is the average of the ensemble variance.

- The member-by-member ("mbm") approach described by Van Schaeybroeck and Vannitsem (2015) is a generalization of (1) to several predictors. The parameter $\gamma$ is made depending on the ensemble spread. The absolute-value spread is chosen: $D_t = \langle \langle |F_t^{m_1} - F_t^{m_2}| \rangle_{m_1} \rangle_{m_2}$. The dependence of $\gamma$ is defined as $\gamma_t = \gamma_1 + \gamma_2 D_t^{-1}$ so that there are four parameters to be optimized. The objective function is the Continuous Ranked Probability Score to be minimized and which can be written as: $CRPS = \langle \langle |F_{C,t}^m - X_t| \rangle_m - D_t/2 \rangle_t$.





### 2.3 Post-processing of precipitation

The lower boundary to zero and the probability of no precipitation have to be taken into account. Here, for three-monthly averages of precipitation forecasts, the linear scaling and the logistic regression are tested. The linear scaling (e.g. Crochemore et al., 2016) is used to correct the raw ensemble members:

$$F_{Ct}^m = \delta F_t^m \tag{3}$$

Three methods to estimate the scaling factor have been tested.

– The scaling factor is obtained by calculating the ratio of the average of the observations and the average of the hindcasts ensemble means ("scr"): $\delta = \langle X_t \rangle_t / \langle \overline{F}_{Ht} \rangle_t$

– The scaling factor is estimated through the minimisation of the CRPS like in the member-by-member approach described above for temperature ("scmc").

– The scaling factor is allowed to vary in time with the absolute-value spread of the ensembles ("scsp"): $\delta_t = \delta_1 + \delta_2 D_t^{-1}$. The two parameters are also estimated by minimisation of the CRPS.

The extended logistic regression ("extlr") relates the probability that the observed precipitation $X_t$ doesn't exceed a threshold $q$ to the predictor $\overline{F}_t$ and to the threshold itself (Wilks, 2009):

$$P\left(X_t \leq q\right) = \left[1 + \exp\left(-f\left(\overline{F}_t\right) - g\left(q\right)\right)\right]^{-1} \tag{4}$$

The methodology described by Roulin and Vannitsem (2012) is followed except that the precipitation forecasts and the thresholds are not power transformed:

$$f\left(\overline{F}_t\right) + g\left(q\right) = \beta_0 + \beta_1 \overline{F}_t + \beta_2 q \tag{5}$$

The three parameters $\beta$ are optimized by maximizing the likelihood function.

### 2.4 Multi-model

The multi-model ensembles have been obtained by simple pooling of all the members of the models involved. All the model combinations have been tested, each over the longest common period. There are 26 combinations possible. Due to the different overlapping periods, the multi-model ensembles (MME) correspond to one of 7 ranges of years among which 4 correspond to a SME (those labelled $b$ to $e$, see Table 1) and 3 correspond to other intersections ($f$ to $h$). The period common to the 5 models, $f$, is only 18 years long i.e. half the length of $a$ (model "A"). The size of the SME vary from 10 to 28 and those of MME vary from 22 to 90. When a post-processing technique is tested on a multi-model ensemble, the hindcasts of the relevant single model ensembles are post-processed separately with the same method. Then the different corrected ensemble are pooled. In the following, multi-models will be designated like: {A,B,C,D} = {A,B,C,D}$_f$ since $f$ is the period common to the four models.





### 2.5 Validation

Three skill scores or relative improvement over a reference unskilled forecast system are estimated. Here the sample climatology is taken as the reference. The first skill score is deterministic: the mean square error skill score (MSSS) compares the forecast ensemble mean with the observation. The reference forecast consist in always issuing the mean of the observations in

the sample. The other two are probabilistic: the Brier skill scores BSS1 and BSS2 where forecast probabilities of some events are compared with their observed occurrence. The events are respectively that the variable will be lower than the lower tercile in the sample distribution (BSS1) and that it will be larger than or equal to the upper tercile (BSS2). The forecast probabilities are taken as the fraction of members satisfying the inequality. The reference forecast is always 1/3. The ranked probability skill score RPSS has also been calculated and is always falling between BSS1 ad BSS2 and is reported at some occasion.

The Pearson correlation coefficient is related to the mean square skill score through the relation (Murphy, 1988; Slater et al., 2017):

$$MSSS = R_{X\overline{F}}^2 - \left(R_{X\overline{F}} - \frac{S_{\overline{F}_t}}{S_{X_t}}\right)^2 - \left(\frac{\langle \overline{F}_t - X_t \rangle_t}{S_{X_t}}\right)^2 = R_{X\overline{F}}^2 - RCB - RUB \tag{6}$$

where the $RCB$ term is a conditional bias and the $RUB$ term is an unconditional bias so that $R^2$ may be considered as the potential skill that might be achieved in the absence of biases (Slater et al., 2017). The unconditional bias is totally removed

with a bias correction based on the full sample. It is easily shown from (2) and (6) that in-sample ("is") post-processing with the variance inflation method results in RCB = 0 and RUB = 0.

The spread-skill relationship is quantified with the Pearson correlation coefficient between the standard deviation of the ensemble members and the mean absolute value of the ensemble-mean forecast error $SPK = R_{S_{t,F^m}, |\overline{F}_t - X_t|}$ (Grimit and Mass, 2007).

Uncertainty about the skill score estimates is evaluated with a bootstrapping procedure. A thousand samples are built by drawing with replacement in the original forecast-observation sample. The skill scores are computed for each of these bootstrap samples and the values are ranked. A forecast will be said to have (potential) skill and be selected if the fifth percentile of the skill score (or the Pearson correlation coefficient) is positive. This percentile method for interval of confidences has been compared with the bias corrected and accelerated method ("BCa", Efron and Tibshirani, 1993) which is more accurate and

palliates to biases arising from non-Gaussian distributions.

As the post-processing and the verification is performed on the same hindcasts, the leave-one-out ("loo") cross-validation scheme is followed (Wilks, 2011). The parameters of a post-processing equation are evaluated on the set of hindcasts and corresponding observations from which the data of one year are put aside. The statistical correction is then applied on the hindcasts of the year left out. The process is repeated each year in turn until the whole sample of post-processed hindcasts and

corresponding independent observations is completed. This scheme applies also to the multi-model ensembles.





## 2.6 Trend

Trends in the time series are computed by least-squares regression and the uncertainty about the slope is evaluated on the base of the same bootstrap samples as above. The de-trending of both the observed time series and the forecast time series prior to the post-processing has been tested either in the in-sample or the leave-one-out cross-validation schemes.

## 2.7 Summary tables

For each variable, there are 6 locations, 4 seasons, 3 lead times, that is 72 *situations* to be forecast. For each situation, we compare the output of 5 models and 26 combinations of them. All model output are post-processed using 3 different methods (4 for precipitation) so that there are 93 (124) forecasts *ensembles* to be evaluated per situation (the raw ensembles are not counted). As the evaluation is performed using 3 skill scores, there is a total of $\sim 2-3\,10^4$ skill score *values* to be analysed.

In the result section, these values will be pooled in various ways. The first way to summarize the results consists in tables – one for every study area and variable – prepared with the following criteria.

- Only the positive values of the skill scores (MSSS, BSS1 and BSS2) selected with the bootstrapping procedure are reported. The Pearson correlation coefficient (R) has been selected the same way but reported as squared values i.e. as potential skill to be compared to MSSS.

- For the single model ensembles (SME), the skill of the three (four) post-processing methods are shown for the model with the highest value (in bold) found among the five models.

- For multi-model ensembles (MME), the value of the skill score of a model combination is larger than that of any of the models available for the range of years and which are post-processed and re-evaluated with the same method and the same period.

- The values of the skill scores of these contributing single models are all larger than 0.1 but not necessarily significantly larger than zero. This threshold is introduced to exclude single model ensembles with poor skill or no skill for the case considered.

- Among the model combinations successful with the above criteria, the MME involving the largest number of single models and resulting in the largest value of the skill score is reported (in bold) along with the skill obtained with the other post-processing methods. Since there is no fair comparison between different MMEs across the different periods, the largest combination of models suited for a specific case is put forward.

## 3 Results and discussion

The skill scores of the seasonal predictions of temperature for the case studies CS3 and CS5, of precipitation for CS1 and CS2 and of mean sea level pressure for CS4 are in Tables 2 to 6. The results for the remaining case studies are in Annex in Tables A1 to A13.



**Table 2.** Skill of seasonal temperature forecasts over the CS3 domain; best single model ensemble (SME) and best multi-model ensemble (MME); post-processing methods: (bc) bias correction, (infl) variance inflation, (mbm) member by member; (-) skill score or R value not significantly positive; the best value in bold.

| Skill | Season | Lead | SME | bc | infl | mbm | MME | bc | infl | mbm |
|---|---|---|---|---|---|---|---|---|---|---|
| $R^2$ | MAM | 0 | A | 0.413 | 0.398 | **0.419** | $\{A,B,C,D\}_f$ | 0.377 | 0.214 | **0.386** |
|  |  | 1 | D | 0.234 | 0.240 | **0.262** | $\{A,D,E\}_d$ | 0.311 | 0.304 | **0.319** |
|  |  | 2 | C | - | - | **0.072** | $\{B,C\}_g$ | - | - | **0.072** |
|  | JJA | 0 | A | **0.124** | - | 0.108 | $\{A,D,E\}_d$ | **0.178** | 0.105 | 0.172 |
|  |  | 1 | D | **0.120** | 0.074 | 0.106 |  |  |  |  |
|  |  | 2 |  |  |  |  | $\{A,D,E\}_d$ | 0.114 | - | **0.117** |
|  | SON | 0 | A | 0.203 | 0.157 | **0.211** | $\{A,D\}_d$ | **0.199** | - | 0.194 |
|  |  | 1 | C | 0.234 | 0.198 | **0.272** | $\{C,E\}_c$ | 0.259 | 0.232 | **0.293** |
|  |  | 2 | A | 0.111 | 0.064 | **0.122** | $\{A,C,E\}_c$ | **0.100** | - | 0.089 |
|  | DJF | 0 | D | 0.188 | 0.185 | **0.215** | $\{A,C,D,E\}_h$ | **0.282** | 0.195 | 0.274 |
|  |  | 1 | C | **0.134** | 0.115 | 0.097 |  |  |  |  |
| $MSSS$ | MAM | 0 | D | 0.401 | 0.389 | **0.416** | $\{A,D\}_d$ | 0.489 | 0.524 | **0.533** |
|  |  | 1 | D | - | - | **0.253** | $\{A,D,E\}_d$ | 0.268 | **0.301** | 0.293 |
|  | SON | 1 | E | - | - | **0.217** |  |  |  |  |
| $BSS1$ | MAM | 0 | D | 0.476 | **0.598** | **0.598** |  |  |  |  |
|  |  | 1 |  |  |  |  | $\{A,D,E\}_d$ | **0.259** | - | - |
|  | JJA | 0 | D | **0.373** | 0.319 | 0.366 | $\{A,D,E\}_d$ | - | - | **0.376** |
| $BSS2$ | MAM | 0 | A | 0.217 | **0.307** | - |  |  |  |  |
|  |  | 1 | E | - | - | **0.218** | $\{D,E\}_d$ | - | - | **0.218** |

At first glance, a small number of cases are emerging for which all three skill scores are simultaneously and significantly positive, be it with different models or model combinations and with different post-processing methods. For temperature:

– for CS1, the spring temperature predicted with a zero lead time;

– for CS2, the spring and autumn temperature with lead time 0;

5 – for CS3, spring temperature with lead time 0 and 1 month;

– for CS4, no case with the three skill score significantly positive;

– for CS5, spring temperature with lead time 0;


**Table 3.** Skill of seasonal temperature forecasts over the CS5 domain; see Table 2.

| Skill | Season | Lead | SME | bc | infl | mbm | MME | bc | infl | mbm |
|---|---|---|---|---|---|---|---|---|---|---|
| $R^2$ | MAM | 0 | B | 0.451 | 0.421 | **0.453** | $\{A,C,D,E\}_h$ | - | **0.335** | - |
| | | 1 | A | **0.346** | 0.310 | 0.338 | $\{B,D,E\}_f$ | **0.216** | - | - |
| | | 2 | B | **0.249** | 0.192 | 0.246 | $\{A,B,E\}_b$ | **0.196** | - | - |
| | JJA | 0 | A | **0.336** | 0.309 | 0.318 | $\{A,B,D,E\}_f$ | 0.259 | 0.201 | **0.266** |
| | | 1 | A | 0.465 | 0.436 | **0.469** | $\{C,D,E\}_h$ | - | - | **0.146** |
| | | 2 | A | **0.282** | 0.263 | 0.269 | $\{D,E\}_d$ | 0.126 | - | **0.139** |
| | SON | 0 | B | **0.130** | 0.074 | 0.114 | | | | |
| | | 1 | | | | | $\{C,D\}_h$ | - | - | **0.112** |
| | DJF | 0 | A | **0.143** | - | 0.115 | $\{A,E\}_e$ | **0.176** | - | 0.120 |
| $MSSS$ | MAM | 0 | B | **0.451** | 0.418 | 0.449 | $\{A,C,D,E\}_h$ | - | 0.329 | **0.363** |
| | | 1 | A | **0.338** | - | **0.338** | | | | |
| | JJA | 0 | A | **0.334** | 0.304 | 0.317 | $\{A,D,E\}_d$ | 0.309 | - | **0.320** |
| | | 1 | A | 0.440 | 0.433 | **0.446** | | | | |
| | | 2 | A | **0.280** | 0.260 | 0.267 | | | | |
| $BSS1$ | MAM | 0 | B | **0.318** | - | - | | | | |
| | | 1 | A | 0.309 | **0.318** | 0.301 | | | | |
| | | 2 | B | - | **0.296** | **0.296** | | | | |
| $BSS2$ | MAM | 0 | A | 0.291 | 0.255 | **0.292** | $\{A,B,D\}_f$ | 0.390 | **0.417** | - |
| | JJA | 0 | A | 0.349 | **0.350** | 0.311 | $\{D,E\}_d$ | - | - | **0.244** |
| | | 1 | A | **0.341** | - | 0.335 | $\{D,E\}_d$ | - | **0.237** | 0.226 |

– for CS6, the summer temperature with lead time 0 to 2 months.

For precipitation, only two cases meet the condition of skilfulness according to the three scores:

– over the CS1 area, winter precipitation with lead time 1 month;

– same for the CS5 area, winter precipitation with lead time 0.

5    As for the mean sea level pressure, there is no case with the three skill scores simultaneously positive. Note however the good skill scores MSSS and BSS2 achieved for the following cases, all at the zero lead time :

– spring mean sea level pressure over the CS2 area;

– spring and winter over the CS4 area, ;



**Table 4.** Skill of forecasts of seasonal precipitation over the CS1 domain; best single model ensemble (SME) and multi-model ensemble (MME); post-processing methods: (scr) rescaling with multiplicative bias, (scmc) rescaling with crps minimisation,(scsp) heteroscedastic rescaling with crps minimisation, (extlr) extended logistic regression; (-) skill score or R value not significantly positive; the best value in bold.

| Skill | Season | Lead | SME | scr | scmc | scsp | extlr | MME | scr | scmc | scsp | extlr |
|---|---|---|---|---|---|---|---|---|---|---|---|---|
| $R^2$ | MAM | 0 | C | 0.221 | **0.242** | 0.181 | 0.231 | $\{B,C,D,E\}_f$ | - | - | **0.248** | - |
| | SON | 0 | B | - | - | - | **0.166** | | | | | |
| | DJF | 0 | C | - | - | **0.100** | - | $\{A,C,D,E\}_h$ | 0.239 | **0.244** | - | - |
| | | 1 | C | 0.156 | 0.190 | **0.568** | 0.198 | | | | | |
| $MSSS$ | MAM | 0 | C | 0.218 | **0.232** | - | - | | | | | |
| | DJF | 1 | C | 0.120 | 0.139 | **0.561** | - | | | | | |
| $BSS1$ | DJF | 1 | C | - | **0.233** | - | - | $\{C,E\}_c$ | - | - | **0.245** | - |
| $BSS2$ | DJF | 1 | C | - | - | **0.583** | - | | | | | |

**Table 5.** Skill of forecasts of seasonal precipitation over the CS5 domain; see Table 4.

| Skill | Season | Lead | SME | scr | scmc | scsp | extlr | MME | scr | scmc | scsp | extlr |
|---|---|---|---|---|---|---|---|---|---|---|---|---|
| $R^2$ | MAM | 0 | C | **0.122** | 0.119 | - | - | $\{A,C,D\}_h$ | - | - | **0.164** | - |
| | | 2 | B | 0.070 | 0.076 | 0.086 | **0.104** | $\{A,B,C\}_g$ | - | - | - | **0.220** |
| | JJA | 0 | D | - | - | **0.113** | - | | | | | |
| | | 1 | D | - | - | **0.112** | - | | | | | |
| | SON | 0 | | | | | | $\{B,C\}_h$ | - | - | - | **0.354** |
| | | 2 | B | - | - | - | **0.194** | | | | | |
| | DJF | 0 | B | 0.217 | 0.161 | - | **0.276** | $\{A,B,D,E\}_f$ | 0.238 | - | - | - |
| | | 1 | | | | | | $\{C,E\}_c$ | - | - | **0.098** | - |
| | | 2 | A | - | - | **0.124** | - | | | | | |
| $MSSS$ | SON | 0 | | | | | | $\{B,C\}_h$ | - | - | - | **0.335** |
| | DJF | 0 | A | 0.227 | **0.231** | 0.212 | - | | | | | |
| $BSS1$ | DJF | 0 | E | 0.259 | **0.263** | - | - | | | | | |
| $BSS2$ | MAM | 2 | B | 0.207 | 0.203 | 0.266 | **0.330** | | | | | |
| | DJF | 0 | A | 0.307 | 0.268 | 0.235 | **0.340** | $\{A,B\}_b$ | - | - | - | **0.296** |





**Table 6.** Skill of seasonal mean sea level pressure forecasts over the CS4 domain; best single model ensemble (SME) and best multi-model ensemble (MME); post-processing methods: (bc) bias correction, (infl) variance inflation, (mbm) member by member; (-) skill score or R value not significantly positive; the best value in bold.

| Skill | Season | Lead | SME | bc | infl | mbm | MME | bc | infl | mbm |
|---|---|---|---|---|---|---|---|---|---|---|
| $R^2$ | MAM | 0 | A | **0.544** | 0.516 | 0.541 | $\{B,C,E\}_g$ | 0.255 | 0.162 | **0.260** |
| | JJA | 0 | A | 0.166 | 0.118 | **0.171** | $\{A,C\}_c$ | 0.388 | - | **0.402** |
| | | 1 | A | **0.112** | - | 0.098 | | | | |
| | | 2 | B | **0.203** | 0.187 | 0.179 | | | | |
| | SON | 0 | A | 0.136 | 0.122 | **0.158** | | | | |
| | DJF | 0 | A | 0.376 | 0.367 | **0.389** | $\{A,D\}_d$ | **0.388** | - | - |
| $MSSS$ | MAM | 0 | A | **0.526** | 0.515 | 0.522 | | | | |
| | DJF | 0 | A | 0.325 | **0.363** | 0.328 | $\{A,C\}_c$ | **0.250** | - | - |
| $BSS2$ | MAM | 0 | C | - | 0.297 | **0.403** | | | | |
| | SON | 0 | A | 0.171 | **0.222** | 0.182 | | | | |
| | DJF | 0 | A | 0.243 | **0.306** | 0.245 | | | | |

– and spring over the CS5 area.

## 3.1 Temperature

Let's pool the 72 forecast situations for the 3 skill scores i.e. a total of 216 cases of different kinds. For temperature, 52 skilful cases are found or almost a quarter of the total. Among these skilful cases, 23 are met with both SME and MME. For 19 cases, only skilful SME are found whereas for the remaining 10 cases, no skilful SME are detected but some combinations of them are. The first benefit of multi-model is therefore to increase the number of cases for which the ensembles of at least one model or one multi-model result in a skill score significantly positive.

The post-processing methods are ranked best differently for SMEs and MMEs. For SMEs, the bias correction, the variance inflation and the member by member methods are ranked first in 19, 11 and 15 cases, respectively (three cases with two ex-aequo). For MMEs, the same methods are ranked first in 5, 8 and 20 cases, respectively. The selection of the post-processing and the model is based on the highest skill score values. The comparison of the distributions of the skill score values of the bootstrap replicates indicates that the differences between the best values and the values ranked second are generally not significant. In some cases, only one post-processing method results in skilful forecasts as for 29% of the SME selected cases and 61% of the MME ones.

In 57% of the SME skilful cases, the model A is chosen. In 82% of the MME skilful cases, the model A is included in the combination. The model D is also included in 82% of the selected MME and model E in 61%. These three models correspond





**Table 7.** Skill of spring temperature forecasts over the CS3 area with model D and lead time 0 month; impact of (Post.) post-processing methods and of (Res.) resampling scheme : (is) in sample, (loo) leave-one-out.

| Post. | Res. | R | $R^2$ | RCB | RUB | MSSS | BSS1 | BSS2 | RPSS |
|-------|------|------|-------|-------|-------|--------|--------|--------|--------|
| raw | | 0.677* | 0.458 | 0.016 | 2.799 | -2.357 | -1.414 | -0.544 | -0.958 |
| bc | is | 0.677* | 0.458 | 0.016 | 0.000 | 0.442* | 0.510* | 0.118 | 0.305* |
| infl | is | 0.677* | 0.458 | 0.000 | 0.000 | 0.458* | 0.650* | 0.080 | 0.351* |
| mbm | is | 0.677* | 0.458 | 0.000 | 0.001 | 0.457* | 0.657* | 0.087 | 0.358* |
| bc | loo | 0.641* | 0.410 | 0.009 | 0.000 | 0.401* | 0.476* | 0.085 | 0.241* |
| infl | loo | 0.626* | 0.392 | 0.003 | 0.000 | 0.389* | 0.598* | 0.032 | 0.302* |
| mbm | loo | 0.646* | 0.417 | 0.001 | 0.000 | 0.416* | 0.598* | -0.007 | 0.281 |

(*) Significantly > 0 at the 5% level.

to three longest hindcast samples (Table 1). The case study CS6 counts the most significantly positive skill score values. Spring and summer are the seasons with the most skilful cases, and, not surprisingly, the lead time 0 is over-represented in the tables. There are more cases with MSSS significantly positive than with BSS2 or with BSS1.

Consider now the 72 situations evaluated with Pearson correlation coefficient R between the ensemble means of a SME or a MME and the corresponding observations. For 53 situations i.e. almost three quarter, R is significantly positive. In Table 7, the correlation coefficient and the skill scores of spring temperature forecast over CS3 with model D and a lead time of 0 month is presented for the raw ensembles and the three post-processing techniques under the in sample or in the leave-one-out settings. The raw ensembles have a significantly negative mean error which translates into a large RUB term in the MSSS decomposition. This unconditional bias is perfectly removed by all post-processing methods. As expected, when using the full

sample, the bias correction is not affecting the RCB term at all. In the same condition, the correlation coefficient is unchanged with the linear transformation of the bc and infl methods and, in the present case, the use of ensemble spread as predictor must not have made any difference in the mbm method. Both the infl and the mbm methods remove completely the conditional biases. Under the leave-one-out re-sampling scheme, the correlation coefficient is decreased to different values for the three methods and the conditional bias of the MSSS decomposition is less perfectly removed by infl and mbm and decreased to the

half for bc. The balance of the three terms determines the actual value of the MSSS and the best skill score value is the mbm's one. In Table 7, the probabilistic scores are reported as well. The raw ensembles have no skill and the in sample values are better than the leave-one-out corresponding ones. The model D has skill at predicting the probability of a seasonal temperature lower than the first tercile in the sample climatology and the best BSS1 is achieved with the mbm method. The RPSS is significantly positive for bc and infl despite that BSS2 is not. The same case with model A results in very similar skill score values except

that BSS2 is skilful (0.307 with infl) and it is one of the very few situation where the same model with the same post-processing method is successful simultaneously for the three skill scores.





### 3.2 Precipitation

The results for precipitation are by far more sparse: on 72 forecast situations, 5 have been forecast with at least one SME and resulted in positive values of MSSS (significant at the 5% level), 3 for BSS1 and 5 for BSS2. All together 13 cases with positive skill scores on a total of 216. An additional two cases correspond to skilled forecasts with some model combinations. Among

the 13 SME cases, the post-processing methods scr, scm, scsp and extlr perform the best in 1, 6, 2 and 4 cases, respectively. The model A and C are at the origin of the majority best skill scores. The case study areas CS2 and CS3 have no skill at all for precipitation. The winter counts 8 skilful cases, spring 3 cases and summer and autumn, one case each.

The best scores for precipitation are obtained with the winter predictions over the area CS1 with the system C and a lead time of one month (Table 4). Complementary results for this case are presented in Table 8. The raw ensembles have a significantly

positive mean error which translates in a large value of the unconditional bias, RUB. For this reason and despite a significantly positive value of the Pearson correlation coefficient, the MSSS of the raw forecasts is much lower than zero. The in-sample configuration allows to see what the post-processing methods are able to achieve. The four post-processing methods perform almost equally perfectly at removing the RUB. Two linear scaling variants (scr and scmc) and the extended logistic regression keep the correlation coefficient unchanged whilst the heteroscedastic scaling scsp has it's value increased (doubled in terms of

$R^2$). For this specific case, it turns out that the absolute error of the ensemble mean is significantly correlated with the standard deviation of the raw ensemble members ($SPK = 0.552$). The conditional bias is largely decreased with the scsp. This post-processing equation allows to condition the correction on the ensemble spread and therefore to simultaneously increase R and reduce RCB. The two other linear scaling variants scr and scmc have fixed coefficient and can just worsen RCB. However, in the in sample configuration, the three scaling methods result in significantly positive MSSS values. The extended logistic

regression is able to reduce RCB and results in a MSSS value larger than those of scr and scmc but not significant. The leave-one-out cross-validation scheme adds a layer of uncertainty. For scr and scmc, it improves the RCB which is slightly diminished relative to the raw ensembles. The correlation coefficient is lower than with the in-sample setting. The only significant MSSS value is obtained with scsd. It is also the only method resulting in a large positive value of BSS2. Note that BSS1 is similar but not significant for the four scaling methods and that RPSS lies somewhere between BSS1 and BSS2.

The very few skilful cases for precipitation seasonal forecasts motivated to address some issues. First, when plotting the bootstrap average versus the values obtained on the full sample, we notice a bias towards low values for the three skill scores (MSSS, BSS1 and BSS2) whereas Pearson correlation coefficient data lie close to the 1:1 diagonal. For a given confidence level, the lower boundary of the confidence interval is probably too small and some cases are incorrectly left out. The percentile method and the BCa method (Efron and Tibshirani, 1993) for confidence intervals have been compared for the 72 forecast

situations with the 5 SMEs and with the 4 post-processing methods. The results (not shown) may be summarized in terms of the number of situations with the correlation coefficient significantly greater than zero. The number of significant cases is larger when the percentile method is used than with the BCa. The same analysis for the three skill scores leads to the opposite conclusion which is consistent with the biases reported above. This is not a proof that the BCa method is appropriate for a specific skill score. A comparison with analytical expressions like in Bradley et al. (2008) is expected to face the same problem





**Table 8.** Skill of winter precipitation over the CS1 area forecast with model C and lead time 1 month; impact of (Post.) post-processing methods and of (Res.) resampling scheme : (is) in sample, (loo) leave-one-out.

| Post. | Res. | R | $R^2$ | RCB | RUB | MSSS | BSS1 | BSS2 | RPSS |
|-------|------|-----|-------|-----|-----|------|------|------|------|
| raw | | 0.552* | 0.305 | 0.065 | 1.287 | -1.047 | -0.265 | -0.470 | -0.374 |
| scr | is | 0.552* | 0.305 | 0.113 | 0.000 | 0.192* | 0.259 | -0.006 | 0.118 |
| scmc | is | 0.552* | 0.305 | 0.111 | 0.001 | 0.193* | 0.249* | -0.014 | 0.109 |
| scsp | is | 0.782* | 0.612 | 0.009 | 0.000 | 0.603* | 0.261 | 0.622* | 0.453* |
| extlr | is | 0.552* | 0.305 | 0.014 | 0.000 | 0.290 | 0.240 | 0.0014 | 0.120 |
| scr | loo | 0.394* | 0.156 | 0.035 | 0.000 | 0.120 | 0.226 | -0.048 | 0.080 |
| scmc | loo | 0.436* | 0.190 | 0.050 | 0.001 | 0.139 | 0.233 | -0.041 | 0.087 |
| scsp | loo | 0.753* | 0.568 | 0.007 | 0.000 | 0.561* | 0.231 | 0.583* | 0.418* |
| extlr | loo | 0.445* | 0.198 | 0.033 | 0.000 | 0.165 | 0.169 | -0.100 | 0.026 |

($^*$) Significantly >0 at the 5% level.

due to for small samples so that large uncertainties subsist. The summary tables for precipitations are based on the use of the BCa confidence intervals for the skill scores and on the percentile confidence intervals for the correlation coefficients.

A second issue is due to the chosen level $\alpha = 0.05$ which allows for some cases to be wrongly selected. The test for SMEs have been re-iterated using $\alpha = 0.005$. For this purpose the number of bootstrap replicates has been increased to 5000. The
number of positive tests (out of 4320) has decreased from 34 to 9 with the more severe level. In terms of the number of forecast situations successful for a skill score with a least one SME, the 13 cases (out of 216) is reduced to 6. Only one situation, the CS1 winter precipitation forecast with the model C and the lead time of one month has two skill scores simultaneously significantly positive. It is worth mentioning that on the one hand, the process of drawing randomly a finite number of bootstrap samples results in an uncertainty on the confidence interval and the outcome of the test may vary from one realization to the other. This
uncertainty could be estimated on the same bootstrap samples. On the other hand, if this number is high and the size of the samples is small, the risk of trivial scores increases. However these problems arise mostly for cases with low skill.

We have used 4 post-processing methods easy to implement. The first two differ by the method of estimating the scale factor. The results are very similar: they are almost always significantly positive together. The skill score for scmc is better than the skill score for scr in more occasions than the reverse in particular when the comparison is made on the BSSs. We would
recommend the scmc based on the minimization of the CRPS. The third scaling method should be investigated more in depth and in more cases. The extended logistic regression is applied in a context different from the known applications on daily data e.g. with respect to sample size. The only four cases where extlr succeeds, it also gives a better value of the skill score than the three scaling methods. As far as the correlation coefficient is concerned, the four post-processing methods scr, scmc, scsp and extlr result respectively in 23, 23, 27 and 20 cases with a significantly positive value. In terms of best values, their number are
respectively 4, 3, 13 and 6 i.e. a total of 26 on a total of 72 forecast situations. Except the cases of scmc like the one described





**Table 9.** Model combination: Pearson correlation coefficient of spring temperature over the CS3 area forecast with lead time 0 month and mbm post-processing; $T$ label for the range of years (see Table 1); A ... E: single model ensembles (ibidem); M4 multi-model ensemble combined from 4 models.

| $T$ | A | B | C | D | E | M4 | M3 | M3 | M2 |
|---|---|---|---|---|---|---|---|---|---|
| $a$ | 0.647* | | | | | | | | |
| $b$ | 0.436* | 0.477* | | | 0.176 | | | | |
| $c$ | 0.459* | | 0.625* | | 0.222 | | | | |
| $d$ | 0.714* | | | 0.646* | 0.311 | | 0.721*[1] | | 0.747*[2] |
| $e$ | 0.645* | | | | 0.317* | | | | |
| $f$ | 0.595* | 0.541* | 0.594* | 0.488* | 0.127 | 0.621*[3] | 0.604*[4] | 0.601*[5] | |
| $g$ | 0.461* | 0.491* | 0.618* | | 0.172 | | | | |
| $h$ | 0.544* | | 0.602* | 0.514* | 0.247 | 0.617*[6] | 0.642*[7] | | 0.609*[8] |

(*) Significantly >0 at the 5% level; multi-models: ([1]) {A,D,E}, ([2]) {A,D}, ([3]) {A,B,C,D}, ([4]) {B,C,D}, ([5]) {A,B,D}, ([6]) {A,C,D,E}, ([7]) {A,C,D}, ([8]) {C,D}.

in Table 8, the goal of post-processing is to keep a correlation as high as the one already existing in the raw ensembles while improving other aspects like the bias, reliability etc.

### 3.3 Model combination

In the setting adopted here, for each case and for a given post-processing method, 26 different combinations of the 5 models are possible and involve 8 different ranges of years. An example is given on Table 9 with the case of spring temperature forecast with a lead time 0 over the CS3 area. The Pearson correlation coefficient is chosen so that more combinations are selected than with a skill score. The mbm post-processing is shown because for this case, this method performs better than the other two. In this table, the rows named as $a$ ... $h$ correspond to the 8 ranges of years and the columns labelled as A ... E correspond to the 5 models so that the diagonal from ($a$,A) down to ($e$,E) shows the values of R of the single models estimated each with its longest sample. The columns headed with M4 to M2 contain the combinations of 4 to 2 models. In this table, only the multi-model results which are positive at the 5% level of significance and greater than all the single model values of the same line are presented in this table. The single model values are not all positive at the same level of significance. For instance, the correlation coefficient of model E satisfies the test only when estimated over the full sample $e$ and the multi-model ensembles {A,D,E} and {A,C,D,E} include the members of E ensembles which are not satisfying the test.

Eight combinations are reported in Table 9. The full five ensemble multi-model is not shown because it scores to 0.595 which is not better than model A. Among the 8, there are two four-ensemble combinations. For the range of years common to the five models ($f$), four models including A stand out with high scores and they combine in the multi-model {A,B,C,D}. This multi-model has been reported in the summary Table 2. However, the difference between this multi-model and the second selected four-ensemble combination is very thin with regard to the variability seen for instance with the scores of a single





model ensemble across the different ranges. The choice of the largest number of models participating to the multi-model is obviously not the only way to follow. We could just take the maximum value of 0.747 with {A,D}. This multi-model and also the second ranked {A,D,E} benefit from the high potential skill of model A for the range of years $d$. As the potential skill of A and of model D vary across the ranges, so must do the potential skill of their multi-model. An additional evaluation of {A,D}

for the range $f$ gives a correlation of 0.611 which is better than any of the five single model ensembles for $f$ but less than 0.621, the correlation for {A,B,C,D}. However, there is no guaranty that the correlation for {A,B,C,D} remains greater than the correlation for {A,D} over $d$ or $a$.

### 3.4   Trends

Trends are first estimated in the longest observation (i.e. ERA Interim) time-series of temperature (sample labelled $a$ in Table

1, 36 years). Warming trends are found in spring temperature in all six domains. For summer and autumn, the trends are also detected except for CS1 and CS2 in summer and CS4 in autumn as for these cases a null trend is not rejected. Concerning winter, no trend is detected in none of the six domains. Altogether, over the 24 cases (6 domains x 4 seasons), 15 have significant trends. When considering a shorter time-series – for instance the sample $b$ with 23 years – the uncertainties about the slope increase and the number of significant trends drops from 15 to 4. For sample $c$ (24 years), this number is 7.

Trends are also estimated in the time-series of seasonal temperature predictions and in particular, in the time-series of the raw ensemble mean at lead-time zero. For model A (on sample $a$), the values of the trends in the forecast time series of the 24 cases are loosely related to the values of the trends in the corresponding observed time series. However, there are 14 cases for which the trends in both the forecast and observed time-series are significant. This is different for model B (sample $b$) as there are 18 cases with significant trends in the forecast time-series against 4 as we have seen in the corresponding observed time-series.

Note that for summer and autumn, the values of the correlation coefficient between the temperature forecast with model B and time (23 years) are greater than those between observed temperature and time over the 36 years – on average 0.702 and 0.421, respectively. The figure is opposite for model C (sample $c$) where 16 cases with no-trend in the forecast time-series agree with the no-trend in the corresponding observed time-series. Model D and E behave like model A.

Now we check the impacts of de-trending both the observed and the forecast time series on the skill scores and compare the

different post-processing methods. Results of MSSS for the 36-years hindcasts of model A and the lead time 0 are presented in Table 10. The trends are estimated "in sample" in order to avoid anticipated additional uncertainties due to cross-validation. Only the 7 cases over 24 for which MSSS without de-trending is significantly positive. All the MSSS values are decreased after the de-trending. For three cases, MSSS values remain significantly positive: the spring temperature in the domains CS1, CS2 and CS3. The four cases with too low or even negative MSSS values correspond to the domains in the Mediterranean region –

CS4, CS5 and CS6 during summer and also CS6 during spring. These four cases have in common larger warming trends than the first group. The bias correction (bc) results in the best scores after the de-trending. The impacts on the other skill scores are no better. From the four cases having skill according to BSS1, two remain significantly positive after de-trending. For BSS2, none of the five skillful cases are left.





**Table 10.** MSSS of seasonal temperature forecast with model A and lead time 0; impact of de-trending; post-processing bc, infl and mbm; trends of the observed and the raw ensemble mean in $^\circ$C year$^{-1}$.

| Domain | Season | bc | infl | mbm | bc | infl | mbm | trend | |
|--------|--------|------|------|------|------|------|------|----------|----------|
|        |        |      |      |      |  —— | de-trended |  —— | observed | forecast |
| CS1 | MAM | 0.601* | 0.586* | 0.466* | 0.552* | 0.532* | 0.451* | 0.037° | 0.034° |
| CS2 | MAM | 0.382* | 0.369* | 0.397* | 0.307* | 0.278* | 0.296* | 0.030° | 0.019° |
| CS3 | MAM | 0.378* | 0.395* | 0.410* | 0.329* | 0.317* | 0.327* | 0.042° | 0.017° |
| CS4 | JJA | 0.352 | 0.406* | 0.343 | 0.111 | 0.231 | 0.075 | 0.049° | 0.052° |
| CS5 | JJA | 0.334* | 0.304* | 0.317* | -0.036 | 0.009 | -0.059 | 0.061° | 0.030° |
| CS6 | MAM | 0.312 | 0.351* | 0.295 | 0.065 | 0.187 | -0.163 | 0.052° | 0.045° |
| CS6 | JJA | 0.531* | 0.522* | 0.539* | 0.161 | 0.151 | 0.138 | 0.071° | 0.046° |

($^*$) Significantly $> 0$ at the 5% level; ($^\circ$) significantly $\neq 0$ at the 5% level.

For temperature, 23-24 years seem not long enough to characterize trends and hence their impact on the skill of seasonal predictions. De-trending both the observed and forecasts time-series reduces drastically the values of the skill scores.

For precipitation, no significant trend is detected anywhere and whatever the length of the time-series with the exception of autumn precipitation increase for CS5 domain and the five periods considered. For mean sea level pressure, some cases with
negative trends are found in the Mediterranean Region for summer and autumn. Further analysis of the EUROSIP hindcasts validation concerning the impacts of sample length and ensemble size are presented in the Appendix B1 and B2

### 3.5 Discussion

One striking outcome of this work is the skill found in the forecasts of spring temperature in all case-studies but one (CS4). Four out of these five case-studies have significant skill according to the three skill scores used in this analysis and, for CS3, the skill
extends even to the lead time of one month. In the literature, most of the reported skills of seasonal predictions over Europe focus on winter and summer when many extremes are likely to occur. Earlier studies describing the results of integrations using observed global sea surface temperature as a lower boundary condition suggested that there was a tendency for the skill of ensemble mean to be highest in the spring season (Brankovic et al., 1994). Speculating why it might be so, these authors noted the potentially conflicting effects of, on the one hand the internal variability which is at a minimum in summer and, on the
other hand, dynamical teleconnections with the tropics usually associated with a winter. As Brankovic et al. (1994) wrote, it is possible that the trade-off between these two requirements occurs in spring. They finally remark that springtime predictability may not be manifest in the fully coupled problem. Using the Arpège model hindcasts of temperature and precipitation from the DEMETER project as forcing data for hydrological seasonal predictions over France, Céron et al. (2010) focus on the predictability of soil moisture and river flows over France for spring because this season covers a large part of the snow
melting period and it is critical for the onset of drought. Singla et al. (2012) extend the study to more rivers and use the





Arpège model hindcasts from the ENSEMBLE project. Alessandri et al. (2011) compared the probabilistic quality of the hindcasts of temperature from both projects. They found that the ENSEMBLE predictions were significantly better than those of DEMETER, in particular for spring in the Euro-Atlantic region. However no skill score was provided from these studies. Note that Doblas-Reyes (2010) uses the persistence of observed anomalies as benchmark. For spring temperature with one-

month lead, persistence has substantial skill (correlation) over central and Northern Europe in spring and the domain with significant skill includes the CS1, CS2 and CS3 area of the present study. According to Doblas-Reyes (2010), the spring temperature forecast with ECMWF seasonal prediction System 3 has skill for CS1 only. Using System 4 (not shown) we have detected significant skill in the three areas like we did with System 5 (SEAS5 or model "A") in Tables 2, A1 and A2.

The second main result is the predictability of summer temperature in the Mediterranean region obtained without de-trending

the observed and forecast data. Significant skill (anomaly correlation) is also shown by Mishra et al. (2018) in their skill assessment of EUROSIP for the forecasts of summer temperature with one-month lead time for areas including the CS4 and CS6 domains of this study for three models. The skill for CS6 is also supported by their results with a probabilistic skill score and for the multi-model. Based on the reliability diagrams, Weisheimer and Palmer (2014) rank 4 to 5 on a scale of 5 the "goodness" of predictions of cold summer or warm summer over the Mediterranean region. The impact of de-trending suggest

that predictability is associated over that region with our ability to make forecast of the climate trend. It seems that no low frequency variability is associated with that predictability.

The configuration of operational multi-model seasonal prediction systems will continue to require specific solutions related to the length of time series and the size of the ensemble. For instance, the Copernicus Climate Change Service (C3S) Seasonal Multi-system - successor of EUROSIP, is currently composed of 5 systems with hindcasts of 24 to 36 year long and ensembles

from 25 to 40 members (https://confluence.ecmwf.int/display/COPSRV/Description+of+the+C3S+seasonal+multi-system).

## 4   Conclusions

The skill of seasonal prediction of near surface atmospheric variables over six case studies in Western Europe and the Mediterranean Region has been investigated using the hindcast data base of the operational EUROSIP multi-model system. The aim was to explore situations defined by the case-study area, the target season and the lead time for which seasonal predictions

would have skill relative to a reference forecast. Three skill scores have been estimated: one deterministic measuring the square of the error of the forecast ensemble mean and two probabilistic based on the forecast probability of higher values than observed on average and probability of lower than average. The hindcasts of five state-of-the-art coupled Atmosphere-Ocean General Circulation Models (GCMs) have been compared to reanalysis data and raingauge data. Different post-processing methods have been compared and the different model combinations have been tested. In order to maximize the opportunities,

the longest hindcast period available have been used for each single model ensemble (SME) and multi-model ensemble (MME) tested. It comes out of this work that hindcasts should be long enough to properly calibrate the post-processing of the seasonal predictions, assess their skill and disentangle the trends and other decadal variations. This echoes e.g. Sansom et al. (2016) and Siegert et al. (2017).





As expected in medium-latitude regions, the skill is poor and difficult to assess. The reasons are the low predictability and the ensemble size but also in the case of retrospective forecasts, the length of the sample which counts from 18 to 36 seasons (years), and, in the case of temperature, the trend. The bootstrap method has been applied to measure the uncertainties and perform tests about whether a skill score is significantly positive (more skill than sample climatology which serves as reference). Not all the results are provided: instead, summary tables contain for every study area, the best scores obtained with a SME or a MME combined with a post-processing algorithm. The most predictable weather element revealed in this study and, to our knowledge, rarely reported, is the spring (average MAM) temperature over the study areas CS1, CS2, CS3 and CS5. For CS3, the skill extends to 1 month lead time (model initiated beginning of February). Autumn temperature over CS2 and summer temperature over CS6 also reveal to have skill. But for the latter, the skill is degraded after de-trending the forecasts and the observations. For precipitation, the skill is sparse: skill for winter precipitation prediction over CS1 achieved mainly by model C and winter precipitation over CS5. As for the mean sea level pressure, spring and winter over the CS4 area are cases with some skill. The progresses of the seasonal predictions of these cases are worth being followed in future.

Within the EUROSIP multi-model seasonal prediction system, all models have skill for a number of situations. For temperature, we have seen that model A counts the most SME's best skill scores and that A, D and E are members of most of best MMEs. For precipitation, model A ranks first in the number of best skill scores followed by model C. In many situations the skill scores obtained after different post-processing methods are of the same order. For temperature, there are more best skill scores after a simple bias correction of SMEs than after applying the variance inflation or the member by member methods. For MMEs, the member-by-member performs best. For precipitation, the re-scaling with a factor estimated by minimizing the CRPS seems the most efficient post-processing method among the four tested. The extended logistic regression is also efficient in a number of situation. The use of the ensemble spread as predictor will be investigated further.

We have found windows of opportunity (diverting the words from Frias et al., 2010) which are heterogeneous because they are obtained by different models or model combinations and post-processing methods. Furthermore, they are pertaining to spatial and temporal averages which are far from requirements of applications like hydrological forecasting. Instead, consistent scenarios of precipitation, temperature and other variables are needed at a finer spatial and temporal scale. Post-processing and downscaling GCM predictions have been the object of intense research to bridge this gap, and the present effort is a step toward the optimal use of operational multi-model systems.



**Table A1.** Skill of seasonal temperature forecasts over the CS1 domain; best single model ensemble (SME) and best multi-model ensemble (MME); post-processing methods: (bc) bias correction, (infl) variance inflation, (mbm) member by member; (-) skill score or R value not significantly positive; the best value in bold.

| Skill | Season | Lead | SME | bc | infl | mbm | MME | bc | infl | mbm |
|-------|--------|------|-----|-----|------|-----|-----|-----|------|-----|
| $R^2$ | MAM | 0 | A | **0.601** | 0.587 | 0.582 | $\{B,C,E\}_g$ | **0.275** | 0.215 | - |
|       |     | 1 | B | **0.299** | 0.264 | 0.225 | $\{A,D,E\}_d$ | **0.209** | 0.143 | - |
|       |     | 2 | B | **0.202** | 0.136 | 0.149 | $\{A,B,E\}_b$ | - | - | **0.162** |
|       | JJA | 0 | A | **0.116** | 0.068 | 0.115 | | | | |
|       | SON | 0 | A | **0.097** | - | 0.081 | | | | |
|       | DJF | 0 | B | **0.149** | 0.096 | 0.099 | $\{B,C,D,E\}_f$ | **0.172** | - | - |
|       |     | 1 | A | 0.160 | **0.194** | 0.113 | | | | |
|       |     | 2 | A | **0.268** | - | - | | | | |
| $MSSS$ | MAM | 0 | A | **0.601** | - | - | $\{A,E\}_e$ | - | - | **0.546** |
| $BSS1$ | MAM | 0 | B | **0.409** | 0.305 | - | | | | |
| $BSS2$ | MAM | 0 | A | **0.267** | - | - | $\{A,D,E\}_g d$ | - | - | **0.375** |

**Appendix A: Skill summary tables**





**Table A2.** Skill of seasonal temperature forecasts over the CS2 domain; see Table A1.

| Skill | Season | Lead | SME | bc | infl | mbm | MME | bc | infl | mbm |
|---|---|---|---|---|---|---|---|---|---|---|
| $R^2$ | MAM | 0 | D | **0.520** | 0.490 | 0.508 | $\{A,C,D,E\}_h$ | - | - | **0.415** |
| | | 1 | D | **0.195** | 0.177 | 0.193 | $\{A,D,E\}_d$ | **0.216** | - | - |
| | | 2 | B | 0.155 | 0.121 | **0.188** | $\{B,C\}_g$ | 0.125 | - | **0.158** |
| | JJA | 0 | D | **0.173** | 0.117 | 0.155 | $\{A,D,E\}_d$ | **0.203** | 0.132 | 0.198 |
| | | 1 | D | **0.121** | - | 0.111 | | | | |
| | | 2 | E | - | - | **0.109** | $\{A,D,E\}_d$ | **0.138** | - | 0.136 |
| | SON | 0 | D | 0.381 | 0.319 | **0.384** | $\{A,D\}_d$ | 0.383 | 0.326 | **0.389** |
| | | 1 | E | **0.282** | 0.257 | **0.282** | $\{A,D,E\}_d$ | 0.284 | - | **0.289** |
| | | 2 | D | **0.165** | 0.120 | 0.155 | $\{A,C,D,E\}_h$ | **0.182** | | 0.181 |
| | DJF | 0 | C | **0.243** | 0.190 | 0.200 | $\{A,B,C,D,E\}_d$ | 0.302 | | **0.324** |
| | | 1 | A | **0.103** | - | - | | | | |
| $MSSS$ | MAM | 0 | D | **0.513** | 0.488 | 0.508 | $\{A,D\}_d$ | 0.520 | 0.511 | **0.541** |
| | SON | 0 | | | | | $\{A,D\}_d$ | 0.383 | - | **0.387** |
| | | 1 | E | **0.280** | - | 0.276 | $\{D,E\}_d$ | - | - | **0.277** |
| | DJF | 0 | | | | | $\{A,B,C,D\}_d$ | 0.292 | - | **0.324** |
| $BSS1$ | MAM | 0 | C | 0.513 | 0.507 | **0.527** | $\{A,D\}_d$ | - | - | **0.407** |
| | SON | 0 | A | **0.347** | 0.301 | 0.310 | | | | |
| | DJF | 0 | | | | | $\{B,C,E\}_g$ | - | - | **0.309** |
| $BSS2$ | MAM | 0 | A | 0.316 | 0.366 | **0.373** | $\{A,D\}_d$ | - | **0.444** | - |
| | SON | 0 | | | | | $\{A,D\}_d$ | - | - | **0.298** |





**Table A3.** Skill of seasonal temperature forecasts over the CS4 domain; see Table A1.

| Skill | Season | Lead | SME | bc | infl | mbm | MME | bc | infl | mbm |
|---|---|---|---|---|---|---|---|---|---|---|
| $R^2$ | MAM | 0 | D | **0.157** | 0.115 | 0.155 | $\{A,E\}_e$ | - | - | **0.088** |
| | JJA | 0 | A | **0.447** | 0.408 | 0.437 | $\{A,C\}_c$ | **0.420** | - - | 0.416 |
| | | 1 | A | 0.207 | 0.166 | **0.213** | $\{A,D,E\}_d$ | **0.176** | - | - |
| | | 2 | A | 0.190 | 0.138 | **0.196** | $\{A,C\}_e$ | **0.170** | - | - |
| | SON | 0 | B | **0.222** | 0.183 | 0.214 | $\{A,B,C\}_f$ | **0.206** | - | - |
| | | 1 | B | 0.174 | 0.120 | **0.182** | | | | |
| | DJF | 0 | A | **0.254** | 0.227 | 0.250 | $\{A,D,E\}_d$ | - | - | **0.212** |
| $MSSS$ | JJA | 0 | A | - | **0.406** | - | | | | |
| | DJF | 0 | A | **0.254** | - | - | | | | |
| $BSS2$ | JJA | 0 | A | - | **0.332** | - | | | | |
| | DJF | 0 | | | | | $\{D,E\}_d$ | - | - | **0.251** |





**Table A4.** Skill of seasonal temperature forecasts over the CS6 domain; see Table A1.

| Skill | Season | Lead | SME | bc | infl | mbm | MME | bc | infl | mbm |
|-------|--------|------|-----|------|------|------|------|------|------|------|
| $R^2$ | MAM | 0 | A | 0.398 | 0.354 | **0.415** | $\{A,B,C,E\}_g$ | - | - | **0.469** |
| | | 1 | A | **0.310** | 0.276 | 0.306 | $\{A,B,C,E\}_g$ | **0.239** | - | - |
| | | 2 | A | **0.191** | 0.145 | 0.171 | $\{A,C,D,E\}_h$ | **0.393** | 0.360 | - |
| | JJA | 0 | B | 0.626 | **0.634** | 0.630 | $\{A,B,C,D,E\}_f$ | - | 0.557 | **0.579** |
| | | 1 | A | 0.540 | 0.528 | **0.549** | $\{A,D,E\}_d$ | 0.521 | - | **0.524** |
| | | 2 | A | **0.439** | 0.423 | 0.437 | $\{A,B,C,E\}_g$ | 0.253 | 0.186 | **0.269** |
| | SON | 0 | E | 0.280 | 0.251 | **0.284** | $\{A,C,E\}_d$ | **0.134** | - | - |
| | | 1 | D | **0.236** | 0.184 | 0.196 | $\{A,D,E\}_d$ | **0.302** | 0.267 | 0.267 |
| | | 2 | D | 0.213 | 0.175 | **0.229** | $\{A,C,D,E\}_h$ | **0.268** | - | - |
| | DJF | 0 | B | **0.169** | 0.077 | 0.103 | $\{A,D,E\}_d$ | **0.159** | 0.091 | - |
| $MSSS$ | MAM | 0 | D | **0.366** | 0.316 | - | $\{A,D\}_d$ | **0.421** | 0.385 | - |
| | JJA | 0 | B | 0.490 | **0.633** | 0.510 | $\{A,B,C,D,E\}_f$ | - | **0.544** | - |
| | | 1 | A | 0.503 | 0.527 | **0.528** | $\{A,B,E\}_b$ | 0.325 | **0.350** | 0.332 |
| | | 2 | A | 0.426 | 0.420 | **0.435** | $\{A,C\}_c$ | - | - | **0.363** |
| | SON | 0 | E | 0.277 | - | **0.281** | | | | |
| | | 1 | | | | | $\{A,D\}_d$ | **0.336** | 0.292 | - |
| | | 2 | D | - | - | **0.228** | $\{A,D,E\}_d$ | 0.297 | - | **0.323** |
| $BSS1$ | MAM | 0 | | | | | $\{A,D,E\}_d$ | **0.351** | - | - |
| | JJA | 0 | C | **0.434** | 0.423 | - | $\{A,C,E\}_c$ | - | - | **0.484** |
| | | 1 | A | 0.395 | **0.428** | 0.418 | | | | |
| | | 2 | A | **0.330** | 0.315 | 0.300 | | | | |
| $BSS2$ | JJA | 0 | E | 0.475 | **0.580** | 0.498 | $\{A,C,D\}_h$ | **0.366** | - | - |
| | | 1 | A | 0.366 | **0.418** | 0.414 | $\{A,E\}_e$ | - | **0.383** | - |
| | | 2 | A | **0.320** | 0.305 | 0.316 | | | | |
| | SON | 2 | A | **0.243** | - | - | | | | |
| | DJF | 0 | | | | | $\{A,D\}_d$ | - | - | **0.281** |
| | | 2 | | | | | $\{A,D,E\}_d$ | - | **0.362** | - |





**Table A5.** Skill of forecasts of seasonal precipitation over the CS2 domain; best single model ensemble (SME) and best multi-model ensemble (MME); post-processing methods: (scr) rescaling with multiplicative bias, (scmc) rescaling with crps minimisation,(scsp) heteroscedastic rescaling with crps minimisation, (extlr) extended logistic regression; (-) skill score or R value not significantly positive; the best value in bold.

| Skill | Season | Lead | SME | scr | scmc | scsp | extlr | MME | scr | scmc | scsp | extlr |
|-------|--------|------|-----|-----|------|------|-------|-----|-----|------|------|-------|
| $R^2$ | MAM | 0 | B | - | - | **0.138** | - | $\{A,B\}_b$ | - | - | **0.173** | - |
| | SON | 1 | | | | | | $\{A,B,D\}_f$ | - | - | **0.190** | - |
| | DJF | 1 | B | - | - | **0.120** | - | $\{B,C\}_g$ | - | - | **0.119** | |

**Table A6.** Skill of forecasts of seasonal precipitation over the CS3 domain; see Table A5.

| Skill | Season | Lead | SME | scr | scmc | scsp | extlr | MME | scr | scmc | scsp | extlr |
|-------|--------|------|-----|-----|------|------|-------|-----|-----|------|------|-------|
| $R^2$ | MAM | 0 | A | **0.123** | 0.117 | - | - | | | | | |
| | JJA | 2 | | | | | | $\{D,E\}_d$ | - | - | **0.085** | - |
| | DJF | 0 | E | - | - | **0.070** | - | $\{A,D\}_d$ | **0.057** | - | - | - |

**Table A7.** Skill of forecasts of seasonal precipitation over the CS4 domain; see Table A5.

| Skill | Season | Lead | SME | scr | scmc | scsp | extlr | MME | scr | scmc | scsp | extlr |
|-------|--------|------|-----|-----|------|------|-------|-----|-----|------|------|-------|
| $R^2$ | MAM | 0 | A | 0.069 | **0.077** | - | - | $\{A,E\}_h$ | **0.084** | - | - | - |
| | JJA | 0 | | | | | | $\{C,D\}_h$ | - | - | **0.091** | - |
| | SON | 0 | A | - | 0.132 | **0.163** | 0.118 | $\{A,C\}_c$ | **0.120** | - | - | - |
| | | 1 | B | | | | **0.155** | $\{B,C,E\}_g$ | - | - | - | **0.154** |
| | | 2 | A | - | - | **0.094** | - | | | | | |
| | DJF | 0 | A | 0.298 | 0.249 | 0.125 | **0.321** | $\{A,C,D\}_h$ | 0.301 | 0.214 | - | **0.416** |
| $MSSS$ | SON | 0 | A | **0.154** | - | - | - | | | | | |
| | DJF | 0 | A | 0.255 | 0.222 | - | **0.304** | $\{A,C,D\}_h$ | - | - | - | **0.397** |
| $BSS1$ | MAM | 0 | A | 0.127 | **0.134** | - | - | | | | | |
| $BSS2$ | DJF | 0 | A | - | 0.150 | - | **0.258** | $\{A,B,C,D\}_f$ | - | - | - | **0.299** |





**Table A8.** Skill of forecasts of seasonal precipitation over the CS6 domain; see Table A5.

| Skill | Season | Lead | SME | scr | scmc | scsp | extlr | MME | scr | scmc | scsp | extlr |
|---|---|---|---|---|---|---|---|---|---|---|---|---|
| $R^2$ | MAM | 0 | D | - | - | **0.125** | - | $\{A,C,D,E\}_h$ | - | - | **0.206** | - |
| | | 2 | C | 0.120 | **0.122** | - | - | $\{C,D,E\}_h$ | 0.186 | **0.203** | - | - |
| | JJA | 0 | B | 0.084 | 0.089 | **0.141** | - | $\{A,C,D,E\}_h$ | 0.161 | **0.174** | 0.169 | - |
| | | 2 | B | 0.153 | 0.156 | **0.212** | - | $\{B,E\}_b$ | - | **0.158** | - | - |
| | SON | 1 | B | **0.103** | 0.085 | 0.094 | 0.067 | $\{B,E\}_b$ | - | - | **0.120** | - |
| | DJF | 0 | A | **0.122** | 0.109 | - | 0.050 | $\{A,E\}_e$ | **0.133** | 0.116 | - | - |
| $MSSS$ | JJA | 0 | | | | | | $\{C,D\}_h$ | 0.205 | **0.209** | - | - |
| $BSS2$ | JJA | 2 | D | 0.222 | **0.259** | 0.192 | - | | | | | |

**Table A9.** Skill of seasonal mean sea level pressure forecasts over the CS1 domain; best single model ensemble (SME) and best multi-model ensemble (MME); post-processing methods: (bc) bias correction, (infl) variance inflation, (mbm) member by member; (-) skill score or R value not significantly positive; the best value in bold.

| Skill | Season | Lead | SME | bc | infl | mbm | MME | bc | infl | mbm |
|---|---|---|---|---|---|---|---|---|---|---|
| $R^2$ | MAM | 0 | A | **0.280** | 0.237 | 0.273 | $\{A,B,D\}_f$ | - | - | **0.225** |
| | | 1 | A | 0.158 | **0.168** | 0.159 | | | | |
| | JJA | 0 | E | - | - | **0.105** | | | | |
| | DJF | 0 | D | 0.165 | **0.169** | 0.164 | | | | |
| | | 1 | C | 0.230 | 0.215 | **0.251** | | | | |
| $BSS1$ | DJF | 0 | E | - | **0.146** | - | | | | |

**Table A10.** Skill of seasonal mean sea level pressure forecasts over the CS2 domain; see Table A9.

| Skill | Season | Lead | SME | bc | infl | mbm | MME | bc | infl | mbm |
|---|---|---|---|---|---|---|---|---|---|---|
| $R^2$ | MAM | 0 | B | 0.314 | 0.282 | **0.322** | $\{A,B,E\}_b$ | 0.382 | 0.321 | **0.383** |
| $MSSS$ | MAM | 0 | | | | | $\{A,B,E\}_b$ | **0.377** | - | **0.377** |
| $BSS2$ | MAM | 0 | B | 0.356 | **0.416** | 0.366 | $\{A,B\}_b$ | 0.370 | - | **0.383** |





**Table A11.** Skill of seasonal mean sea level pressure forecasts over the CS3 domain; see Table A9.

| Skill | Season | Lead | SME | bc | infl | mbm | MME | bc | infl | mbm |
|-------|--------|------|-----|-----|------|-----|-----|-----|------|-----|
| $R^2$ | MAM | 0 | B | 0.296 | 0.272 | **0.298** | $\{B,E\}_b$ | **0.321** | 0.283 | **0.321** |

**Table A12.** Skill of seasonal mean sea level pressure forecasts over the CS5 domain; see Table A9.

| Skill | Season | Lead | SME | bc | infl | mbm | MME | bc | infl | mbm |
|-------|--------|------|-----|-----|------|-----|-----|-----|------|-----|
| $R^2$ | MAM | 0 | A | **0.365** | 0.326 | 0.359 | $\{A,C,E\}_c$ | 0.394 | 0.317 | **0.408** |
| | JJA | 0 | A | **0.224** | 0.184 | 0.212 | $\{A,B\}_b$ | **0.262** | - | 0.256 |
| | | 1 | A | 0.149 | 0.089 | **0.154** | $\{A,B,C\}_g$ | **0.139** | - | 0.137 |
| | | 2 | B | **0.196** | 0.159 | 0.188 | | | | |
| | DJF | 0 | | | | | $\{A,B,E\}_b$ | 0.134 | - | **0.145** |
| $MSSS$ | MAM | 0 | A | **0.365** | - | 0.358 | $\{A,C\}_c$ | - | **0.327** | - |
| $BSS2$ | MAM | 0 | A | - | - | **0.324** | | | | |

**Table A13.** Skill of seasonal mean sea level pressure forecasts over the CS6 domain; see Table A9.

| Skill | Season | Lead | SME | bc | infl | mbm | MME | bc | infl | mbm |
|-------|--------|------|-----|-----|------|-----|-----|-----|------|-----|
| $R^2$ | MAM | 0 | A | 0.253 | 0.182 | **0.254** | $\{A,B,C,E\}_g$ | 0.140 | - | **0.152** |
| | JJA | 0 | A | **0.125** | - | 0.122 | $\{A,E\}_e$ | - | - | **0.120** |
| | | 1 | E | **0.169** | 0.108 | 0.168 | $\{C,E\}_c$ | **0.112** | - | 0.110 |
| | | 2 | A | **0.089** | - | 0.088 | $\{A,E\}_e$ | **0.112** | - | 0.107 |
| | SON | 0 | A | 0.093 | - | **0.103** | | | | |
| | | 1 | A | - | - | **0.085** | | | | |
| $BSS2$ | JJA | 1 | C | - | - | **0.236** | | | | |





**Table B1.** Seasonal prediction of temperature; number of skilful cases (at 5% level) for the single models A and D and for the multi-model {A,D} evaluated over the ranges $d$ and $f$; (NMM) number of situations for which a multi-model is scoring better than the single models that make it up; (NBS) number of situations for which a multi-model covering one range is scoring better than that covering the other range; skill scores of the 72 situations and post-processing with mbm.

|      | $A_d$ | $D_d$ | $\{A,D\}_d$ | $A_f$ | $D_f$ | $\{A,D\}_f$ | $NMM_d$ | $NMM_f$ | $NBS_d$ | $NBS_f$ |
|------|-------|-------|-------------|-------|-------|-------------|---------|---------|---------|---------|
| R    | 33    | 33    | 35          | 20    | 13    | 20          | 19      | 11      | 15      | 7       |
| MSSS | 9     | 5     | 11          | 1     | 0     | 3           | 7       | 3       | 6       | 2       |
| BSS1 | 2     | 2     | 4           | 2     | 0     | 2           | 2       | 0       | 2       | 0       |
| BSS2 | 5     | 0     | 9           | 0     | 1     | 0           | 4       | 0       | 4       | 0       |

## Appendix B:  Sample size and ensemble size

### B1   Sample size

The multi-model {A,D} is analysed further to gauge the impact of the sample size. For this purpose, the hindcasts of models A and D already post-processed over the range $f$ (18 years) have been pooled to build $\{A,D\}_f$. This multi-model is compared

with $\{A,D\}_d$ over the range $d$ (29 years) and to the respective SMEs ($A_f$, $D_f$, $A_d$ and $D_d$). The member by member post-processing method is applied to the seasonal predictions of temperature and all the 72 forecast situations are considered (6 domains x 4 seasons x 3 lead times). The results are summarised in Table B1 in terms of numbers of skilful forecasts and numbers of best scoring multi-models.

The models A and B and the multi-model {A,D} have their forecasts positively correlated with the observations in nearly

the half of the 72 situations when the samples correspond to the range $d$. This number falls to less than one third when the samples correspond to the shorter range $f$. Out of these cases with positive correlation for {A,D}, about one half have a better correlation than the single models separately – that makes 19 cases for $\{A,D\}_d$ and 11 for $\{A,D\}_f$. Comparing the values of the correlation coefficient for these successful multi-models, for 15 cases, the multi-model over $d$ have a better correlation than that over $f$, and for 7, it is the reverse. Therefore taking the largest common sample is advantageous at the different stages

of the model selection. This choice could also lead to a decrease in the uncertainty about the correlation coefficient as the comparison of the confidence intervals seems to reveal. Here the width of the confidence interval is defined as a difference of percentiles in the ranked bootstrap replicates (P95-P05). For 11 cases out of 35 with a positive correlation for $\{A,D\}_d$ (Table B1), the confidence interval of the multi-model is narrower than those of the single models separately. For $\{A,D\}_f$, this number is 7. For 10 cases, the confidence intervals of $\{A,D\}_d$ are even narrower than those of the corresponding $\{A,D\}_f$ against only

one case for the reverse.

When it comes to making the same analysis on skill scores, we must interpret the small numbers with caution. With the larger sample $d$, the number of skilful cases according to MSSS is 9 and 5 for model $A_d$ and $D_d$, respectively (Table B1). This qualification of skilfulness is based on the 5% level so that on a set of 72 forecast situations, it might happen that, on average,





**Table B2.** Seasonal prediction of temperature with model B using hindcast S13 (M=28 members) and pooled S12+S13 (M=40 members) ; Pearson correlation coefficient for the 72 situations : (a) the number of selected cases (significant at 5% level), (b) comparison between the two datasets (M) of the number of cases with the best value and (c) of the number of cases with the narrowest confidence interval (P95-P05 of bootstrap replicates of R), and (d) comparison of between the six combinations of the number of best performing post-processing method (pp).

|  | (a) | # | R>0* | (b) | # max | R(M) | (c) | # min | CI(M) | (d) | # max | R(pp) |
|---|---|---|---|---|---|---|---|---|---|---|---|---|
|  | bc | infl | mbm | bc | infl | mbm | bc | infl | mbm | bc | infl | mbm |
| S13 | 24 | 19 | 22 | 5 | 4 | 7 | 9 | 10 | 9 | 4 | 0 | 3 |
| S12+S13 | 26 | 22 | 26 | 21 | 19 | 18 | 16 | 12 | 17 | 11 | 1 | 6 |

3.6 be selected by chance – being understood that in the present case it is a mixture of cases of various kinds. For the multi-model $\{A,D\}_d$, there are 11 skilful cases among which 7 with better MSSS values than the corresponding single models and 6 with confidence intervals narrower than those of the single models. With the smaller sample $f$, there are only 3 skillful cases for the multi-model. For these 3 cases, the multi-model is scoring better than the single models and has a narrower confidence

interval. In conclusion, there are some clues that more cases are selected by taking the longest hindcast time-series available and pooling model ensembles.

## B2   Ensemble size

The 28-member ensembles of the release S13 of the model B hindcasts (see Section 2.1) are compared with the 40-member pooled ensembles S12+S13 . For these two datasets, the seasonal temperature forecasts corresponding to the 72 situations and

post-processed with the three methods are evaluated with the Pearson correlation coefficient and the results are summarised in Table B2. The number of cases for which the forecasts are significantly positively correlated with the reanalysis data is of the same order for the two ensemble sizes and the three post-processing methods with a little smaller number for S13 and for the variance inflation (Table B2a). For all three post-processing methods, there are more cases with a better correlation for the pooled ensembles (Table B2b) and more cases with less uncertainty in the value of the correlation coefficient for the pooled

ensembles than for the S13 ensembles (Table B2c). On a total of 25 cases (over 72) with at least one forecast with a significant correlation (out of six for 2 sizes and 3 post-processing methods), 11 best values are obtained with the pooled ensemble and the simple bias correction (Table B2d). The variance inflation method has only one case for which this method is best performing (ibidem). The results for the skill scores are not shown: there are only two cases with skill for MSSS, three for BSS1, and one case for BSS2. Some of these results are included in the summary tables (Tables A1 and 3). In this analysis, the most striking

effect of ensemble size is the larger number of cases with the best correlation for the pooled ensembles.





*Author contributions.* Both authors have designed the analysis performed in this paper. ER implemented the postprocessing techniques, the evaluation tools and made the computations. Both authors have contributed to the interpretation of the results and to the writing of the manuscript.

*Competing interests.* The authors declare that they have no conflict of interest.

5  *Acknowledgements.* This work contributes to the MEDSCOPE project (ERA4CS).





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
