# Peer review of "Post-processing of seasonal predictions – Case studies using the EUROSIP hindcast data base"

_Nonlinear Processes in Geophysics, 2019_

## Referee Comment (RC1) · Anonymous Referee #1 · 21 Oct 2019

The authors compare several simple post-processing methods for seasonal forecasts of temperature, pressure and precipitation from five models in the EUROSIP ensemble forecasting system. Results are presented for six regions of southern and western Europe, at lead times of between zero and two months. Both deterministic and probabilistic forecast skill is assessed relative to climatology. Results are presented for both individual models and various combinations of the EUROSIP models.

I am unclear what the purpose of this study is. A large amount of effort has been expended to perform a huge number of comparisons. However, the study lacks sufficient structure to provide useful, original, generalizable results regarding either the

best post-processing methods, the best models, useable lead times, useful ensemble sizes or required training period. In its current form this manuscript feels more like an internal technical report than a piece of research of international interest.

The authors are clearly motivated by the two hydrological applications described in Section 2.1, but the inclusion of the other four study areas seems unnecessary and makes the manuscript extremely long. The additional study areas do not appear to be chosen to be representative of difference climate regimes or other systematic differences and no reference is made to their relative positions or conditions in the text.

The introduction mentions a lot of details relevant for post-processing studies, but at the same time is often disjointed and lacks context, jumping between topics in the same paragraph. The authors also specifically mention some previous findings and recommendations which they later ignore, e.g., Page 3, Lines 5-7.

The notation used throughout Section 2 to describe difference post-processing methods is never explained, making it difficult to be certain of what is being proposed.

Most of the results presented are based on in-sample comparisons which would never be used in practice. Why not just show the cross-validated results which are more believable?

The authors often attempt to summarize the results by the number of scenarios where a given post-processing method performed best. This seems potentially problematic and misleading given the small number of regions considered, and the fact the results are pooled across regions, seasons, lead times, skill scores etc.

On Page 14, the authors appear advocate choosing the method of confidence interval calculation based on which one gives most significant results rather than any scientific basis!

Finally, in Section 3.4 we are told that almost all the positive results described over the last eight pages were simply due to warming trends.
In summary, I find no evidence of original results that would be of interest to a wider audience.

---

## Referee Comment (RC2) · Anonymous Referee #2 · 16 Feb 2020

Review of the manuscript "Post-processing of seasonal predictions - Case studies using the EUROSIP hindcast data base" by Emmanuel Roulin and Stéphane Vannitsem

The authors evaluate the performance of the seasonal hindcasts from EUROSIP multi-model system for a number of variables and regions in Europe, and consider various post-processing techniques and various multi-model combinations.

I was overwhelmed with the sheer amount of presented information but I don't feel I learned anything valuable from the study. I don't exactly know what the goal of the study is. If it is to demonstrate that some regions in Europe have some skill on seasonal time scales, then I don't think there is really need to consider all these multitude of

the results (2-3e4 in the author's estimation). This particular point could have been demonstrated with a very small subset of the model results. On the other hand, if the point of the paper to demonstrate advantages or disadvantages of a particular post-processing method, then after reading the manuscript I still have any idea which method is recommended. It is also not clear what multi-model combination is to be used. It is all look rather random and disorganized.

A sheer amount of the considered statistics presents a selection problem. Even if there were no skill in none of the model predictions, some of the skill estimates will be positive just by chance due to sampling variability. Given the large amount of statistics considered ($\sim$2-3e4), it is not hard to imagine that many hindcasts will appear skilful, even in the absence of true skill. Many if not all of the skill estimates summarized in Tables are likely to be inflated and are not accurate representations of the true skill.

I don't believe it is sufficient for a scientific paper just to give a description of the results, without providing any useful insight. No new post-processing methodology is suggested. No new scientific insights are gained. Language needs some polishing as well.

I can't recommend this paper for publication.

---

## Author Comment (AC1) · 8 Jul 2020

Author's response to Reviewer 1

R1: "The authors compare several simple post-processing methods for seasonal forecasts of temperature, pressure and precipitation from five models in the EUROSIP ensemble forecasting system. Results are presented for six regions of southern and western Europe, at lead times of between zero and two months. Both deterministic and probabilistic forecast skill is assessed relative to climatology. Results are presented for both individual models and various combinations of the EUROSIP models.

I am unclear what the purpose of this study is. A large amount of effort has been expended to perform a huge number of comparisons. However, the study lacks sufficient structure to provide useful, original, generalizable results regarding either the best post-processing methods, the best models, useable lead times, useful ensemble sizes or required training period. In its current form this manuscript feels more like an internal technical report than a piece of research of international interest. "

A: The point is that there is no "generalizable" result. One of the messages of this paper to the international community is that a post-processing method has to be tailored for a specific local application taking advantage of the most relevant information.

R1: "The authors are clearly motivated by the two hydrological applications described in Section 2.1, but the inclusion of the other four study areas seems unnecessary and makes the manuscript extremely long. The additional study areas do not appear to be chosen to be representative of difference climate regimes or other systematic differences and no reference is made to their relative positions or conditions in the text."

A: The six study areas appeared as a manageable number to explore a bit of variability in the neighborhood of the two river basins. The first group is aligned on a transect South-South-West to North-North-Est, not directly exposed to the Atlantic Ocean or North Sea and with contrasted latitude - plus Great-Britain as a closer comparison. The second transect follows the same parallel as the Greek river basin from west of the Mediterranean Sea to east of it.

We agree that the inclusion of additional four study areas makes the manuscript extremely long. This also the feeling of reviewer 2. The reviewer 1 is right in noticing that we didn't make reference to their relative positions and conditions in the text. Since this study isn't aimed at covering all European countries, we now focus on CS3 and CS5 and skip the other four study areas. We also restrict the analysis to temperature and precipitation and leave aside the mean sea level pressure.

R1: "The introduction mentions a lot of details relevant for post-processing studies, but at the same time is often disjointed and lacks context, jumping between topics in the same paragraph. The authors also specifically mention some previous findings and recommendations which they later ignore, e.g., Page 3, Lines 5-7."

A: The view point of the present work differs from Hagedorn et al (2005) as we are interested in specific regions one at a time and not in the global scale.

The post-processing involves many aspects which are difficult to isolate. The introduction section will be re-worked, in particular to improve the linking and the context.

R1: "The notation used throughout Section 2 to describe difference post-processing methods is never explained, making it difficult to be certain of what is being proposed."

A: We will re-work the Section 2 and complete missing notation explanations.

R1: "Most of the results presented are based on in-sample comparisons which would never be used in practice. Why not just show the cross-validated results which are more believable?"

A: This is not correct. Only two tables (Table 7 and Table 8) compare results based on raw forecasts, forecasts post-processed in sample and forecasts post-processed with cross-validation. In sample results have been included to put into evidence what a post-processing method is aimed to compared to what is achieved in practice given the sample available. The remaining 22 tables (Tables 2-6, 9,10, A1-A13, B1 and B2) all show exclusively cross-validated results.

R1: "The authors often attempt to summarize the results by the number of scenarios where a given post-processing method performed best. This seems potentially problematic and misleading given the small number of regions considered, and the fact the results are pooled across regions, seasons, lead times, skill scores etc."

A: We warned the reader about the limitations in doing so. As the number of regions has been further decreased, we avoid such attempts at summarizing. We simply underpin the few significant results with an estimate of the accuracy of the confidence intervals themselves (see below).

R1: "On Page 14, the authors appear advocate choosing the method of confidence interval calculation based on which one gives most significant results rather than any scientific basis!"

A: We acknowledge that this part is ambiguous. To solve this problem, we have undertaken a fast survey of alternative confidence intervals techniques and also of methods to assess their accuracy. We performed a number of Monte Carlo simulations to sort out the techniques best suited for each skill score. We started with the percentile and BCa bootstrap methods already used for the first manuscript. It appeared that none provided accurate confidence intervals. Therefore, we included several improved techniques by using the toolbox "ibootci" (Andrew C. Penn, 2019). Finally, we developed a parametric bootstrap for the Brier Skill Score which was the hardest skill score to deal with given the small samples. This additional piece of research brings new insights to the analysis of seasonal predictions and motivates us to brave the final sentences of both reviewers.

R1: "Finally, in Section 3.4 we are told that almost all the positive results described over the last eight pages were simply due to warming trends."

A: This is a terse way of presenting the impact of trends. We could argue that the ability of a seasonal forecast system at correctly predicting both the trend and a signal about the trend should be included in the system's specifications.

R1: "In summary, I find no evidence of original results that would be of interest to a wider audience."

A: The study on the accuracy of the confidence intervals completes the analyze the seasonal predictions together with the multi-systems combination and the post-processing techniques. This study was triggered by one remark from Reviewer 1 and

it enhances significantly the results. We are confident that Reviewer 1 will agree that the interest to a wider audience has increased accordingly.

Author's response to Reviewer 2

R2: "The authors evaluate the performance of the seasonal hindcasts from EUROSIP multi-model system for a number of variables and regions in Europe, and consider various post-processing techniques and various multi-model combinations.

I was overwhelmed with the sheer amount of presented information but I don't feel I learned anything valuable from the study. I don't exactly know what the goal of the study is. If it is to demonstrate that some regions in Europe have some skill on seasonal time scales, then I don't think there is really need to consider all these multitude of the results (2-3e4 in the author's estimation). This particular point could have been demonstrated with a very small subset of the model results. On the other hand, if the point of the paper to demonstrate advantages or disadvantages of a particular post-processing method, then after reading the manuscript I still have any idea which method is recommended. It is also not clear what multi-model combination is to be used. It is all look rather random and disorganized."

A: As explained in the response to the first reviewer, the sheer amount of presented information has been drastically reduced by focusing on the two areas that motivated this research instead of six areas, and by dropping the analysis of mean sea level pressure which didn't yield as much as expected. This allows us to perform "case studies" about a limited number of topics related to the seasonal predictions of meteorological variable and not an in-between extensive and exhaustive verification analysis of a prediction system. So, this revised manuscript is clearly not aimed at demonstrating that some regions in Europe have some skill on seasonal time scales. We put ourselves and the reader in the place of a local end user who wants to know what information seasonal predictions can be delivered about his/her river basin and, in particular, about the following questions: is the average temperature of the following season going to be

warmer than the average? Is it going to be colder? Shall it rain more or less than usual? The total depth of precipitation is also a relevant result to look at by the local water manager. How reliable are these predictions? The answers are given as skill scores relative to the climatology.

On the other side, the various post-processing techniques, the various multi-model combinations and the various bootstrap methods to build confidence intervals associated to the skill scores are the three axes along which we want to analyze the EUROSIP hindcasts (see an example in Figure 1 for two models, two post-processing techniques, and five confidence interval methods). The goal is less to recommend one post-processing technique in particular or the best multi-model combination than showing how these aspects interact in the harsh constraint of small samples.

However, some novelties are proposed: among the post-processing techniques tested, some are new and some others were never used before in the context of seasonal forecasting. The multi-model combinations are characterized not only by the models included but also by the sample size which is taken as large as permitted by the corresponding hindcasts. Usually, only the hindcast period common to all models is taken into account. A new parametric bootstrap method inspired by Murphy and Wilks (1998) has been designed for the confidence intervals of the Brier Skill Score and its accuracy compared with the accuracy of a bunch of classical methods.

R2: "A sheer amount of the considered statistics presents a selection problem. Even if there were no skill in none of the model predictions, some of the skill estimates will be positive just by chance due to sampling variability. Given the large amount of statistics considered (2-3e4), it is not hard to imagine that many hindcasts will appear skilful, even in the absence of true skill. Many if not all of the skill estimates summarized in Tables are likely to be inflated and are not accurate representations of the true skill. I don't believe it is sufficient for a scientific paper just to give a description of the results, without providing any useful insight."

The true skill has been our main concern since the very first positive correlation co-efficient detected and the reading of, e.g. Kumar (2009). Monte Carlo simulations provided some information about the accuracy of the confidence intervals on which we base the selection of "significantly positive skill scores". For the Brier Skill Score, we started with the same experiment as described by Bradley et al. (2008) for their ana-lytical expression of the sampling variance and we compared some of their results with those obtained with bootstrapping. Then we focused on our test cases to select the most appropriate confidence intervals for seasonal predictions. For the Mean Square Skill Score, we first investigated bivariate normal distributions and then looked at non-normal samples (Bishara and Hittner, 2017) in the range of the samples retrieved in the EUROSIP hindcasts data-base. This analyze is shortly reported in the revised manuscript because it seems to be lacking despite the frequent use of bootstrapping in similar studies.

R2: "No new post-processing methodology is suggested. No new scientific insights are gained. "

A: New post-processing methodology: see above. New scientific insight: confidence intervals methods best suited for each skill score. New parametric bootstrap method for the Brier Skill Score. Thorough scanning of the multi-model combinations taking into account the maximum information available. Limited skill over the two study areas: spring temperature over Belgium and winter precipitation over Greece.

R2: "Language needs some polishing as well."

A: The revised manuscript will be read by a native speaker.

R2: "I can't recommend this paper for publication."

A: The excess information pointed in the manuscript dieted significantly. We end up with a consistent set of results showing how to take the maximum advantage of the hindcasts of a multi-system for seasonal predictions like EUROSIP. The sample variability has been taken into account and careless conclusions avoided. We have identified a number of novelties and insights which might be worth being shared.

References

Bishara, A. J., and J. B. Hittner: Confidence intervals for correlations when data are not normal. Behav. Res., 49, 294-309, doi:10.3758/s13428-016-0702-8, 2017.

Bradley, A. A., S. S. Schwartz, and T. Hashino: Sampling Uncertainty and Confidence Intervals for the Brier Score and Brier Skill Score. Wea. Forecasting, 23, 992–1006, https://doi.org/10.1175/2007WAF2007049.1., 2008.

Efron, Bradley, and Robert J. Tibshirani: An introduction to the bootstrap. Chapman & Hall, New York, London, 436 pp., 1993.

Kumar, A.: Finite Samples and Uncertainty Estimates for Skill Measures for Seasonal Prediction. Mon. Wea. Rev., 137, 2622–2631, https://doi.org/10.1175, 2009.

Murphy, A. H., and D. S. Wilks: A Case Study of the Use of Statistical Models in Forecast Verification: Precipitation Probability Forecasts. Wea. Forecasting, 13, 795–810, https://doi.org/10.1175/1520-0434(1998)013<0795:ACSOTU>2.0.CO;2, 1998.

Penn, Andrew C.: Ibootci, GitHub, https://www.github.com/acp29/iboot, 2019.

[Figure]

[Figure]

[Figure]

**Fig. 1.** Brier Skill Score of the probability of spring temperature over Belgium being lower than the first tercile predicted with a lead time zero. Confidence intervals: comparison of five bootstrap methods.

---

## Editor Comment (EC1) · Daniel S. Wilks (Editor) · 1 Oct 2020

I think the paper still has serious problems, and is not acceptable in current form. Most problematic is the selective presentation of results, for (one assumes) only seasons that show best results. No attempt has been made to account for multiple testing in evaluation of the "significant" results among many evaluations, even for the presented results. This is so even though the Wilks (2016) False Discovery Rate paper is cited in the introduction on page 3. This concern was also voiced by Reviewer 2 of the original submission. Consequently, meaningful forecast skill has been demonstrated.

[Figure]

2019-45, 2019.